# Using Trait-Based Methods to Study the Response of Grassland to Fertilization in the Grassland in Semiarid Areas in the Loess Plateau of China

**DOI:** 10.3390/plants11152045

**Published:** 2022-08-04

**Authors:** Yuting Yang, Zhifei Chen, Bingcheng Xu, Jiaqi Wei, Xiaoxu Zhu, Hongbin Yao, Zhongming Wen

**Affiliations:** 1College of Grassland Agriculture, Northwest A&F University, Yangling 712100, China; 2College of Life Sciences, Guizhou University, Guiyang 550025, China; 3Institute of Soil and Water Conservation, Northwest A&F University, Yangling 712100, China

**Keywords:** functional traits, productivity, grassland, fertilization, Loess Plateau

## Abstract

Grassland is the dominant vegetation type in the Loess Plateau, and grassland productivity and processes are limited by nitrogen (N) and phosphorus (P). Studies have shown that productivity would change following fertilization in the grassland. The response of productivity to fertilization mainly depends on the dominant species traits. Trait-based methods provide a useful tool for explaining the variations in grassland productivity following fertilization. However, the relative contribution of plant functional traits to grassland productivity under N and P addition in the Loess Plateau is not clear. We measured aboveground biomass (AGB) and leaf N content (LN), leaf P content (LP), leaf N/P ratio (LN/P), specific leaf area (SLA), leaf tissue density (LTD), leaf dry matter content (LDMC), and maximum plant height (H_max_) to study how these plant functional traits regulate the relative biomass of different species and grassland productivity following fertilization. Our results showed, that under different nutrient addition levels, the linkages between plant functional traits and the relative biomass of different species were different. Community AGB was positively related to community−weighted mean LN (CWM_LN), CWM_LN/P, CWM_SLA, and CWM_H_max_, but negatively related to CWM_LTD and CWM_LDMC. Dominant species traits largely determined grassland productivity, in line with the mass ratio hypothesis. These findings further highlight the close linkages between community-level functional traits and grassland productivity. Our study contributes to the mechanisms underlying biodiversity–ecosystem function relationships and has significance for guiding semiarid grassland management.

## 1. Introduction

Anthropogenic drivers of environmental changes (e.g., fertilization) will cause changes in biodiversity and ecosystem function [1]. Recent studies have also shown that the decline in biodiversity may change ecosystem functioning [2,3]. In the context of global change, studies on biodiversity and ecosystem function are increasing [4,5]. Grassland is one of the largest ecosystems in the world [6]. Thus, it is necessary to study the relationship between biodiversity and grassland ecosystem productivity under environmental change. In most previous field experiments, the effect of fertilization on productivity was mainly dependent on the species, and species diversity was often used to explain the changes in productivity following fertilization [1,7,8]. However, it is being recognized more and more that functional diversity rather than species diversity finally drives the biodiversity–ecosystem function relationship [9]. Furthermore, plant functional traits can better help to explain and understand ecosystem function than species-based metrics (e.g., species richness and species abundance) [10,11].

Plant functional traits are defined as plant physiological and morphological characteristics that influence plant growth, survival, and so on [12]. They are closely related to the community productivity or plant adaptation [13,14]. Trait-based methods have become a useful method to understand ecosystem function [15,16]. The literature has shown that plant functional traits can modulate grassland productivity along nutrient gradients [17]. The leaf is the main organ of photosynthesis in plants, and has a significant impact on ecosystem function [18,19], such as specific leaf area (SLA), which represents changes in the leaf economics spectrum, indicating the ability of species to respond to rapid growth [18]. In view of this, understanding the linkages between plant functional traits and productivity under nitrogen (N) and phosphorus (P) addition in the Loess Plateau is of increasing importance for guiding grassland management and restoration.

The mass ratio hypothesis suggests that the ecosystem process is determined to a great extent by the dominant species traits, and community productivity is mainly determined by the mean functional traits of plant species [20,21]. An increasing body of studies suggested that the traits of dominant species related to competition play key roles in the response of the community to environment [22]. The mass ratio hypothesis used community−weighted mean (CWM) traits to characterize plant functional traits [21], and the CWM traits of these species can describe the community property accurately [23]. For example, rapid growing plant species from nutrient-abundant environments usually have high SLA and leaf N content (LN) and low leaf dry matter content (LDMC), but the opposite traits are found in species from nutrient-poor environments [24,25]. Studies have shown that SLA affects community productivity by affecting the maximum photosynthetic rate and increasing light interception [26]. Some studies also showed that plant nutrient characteristics would influence productivity [17]. Thus, the challenge is to identify the key plant functional traits of dominant species that have important effects on community aboveground biomass (community AGB) under N and P addition.

N addition would influence community productivity, but it will also aggravate P limitation [27]. As in the case of the Loess Plateau, grassland is the main vegetation type in the region [28,29]. However, the grassland productivity here is usually limited by both N and P in the Loess Plateau [30,31]. Thus, if we only add N, it will not be beneficial to the restoration of the grassland in the Loess Plateau. Accordingly, it is necessary to apply N and P together to promote grassland restoration. A better understanding of mechanisms affecting ecosystem function under N and P addition is essential for guiding grassland management. However, to date, few studies have investigated the linkages between plant functional traits and grassland productivity under both N and P addition in the Loess Plateau.

Accordingly, in this study, we designed a four-year N and P addition experiment to assess which functional traits have an important influence on community AGB and how dominant species traits affect ecosystem function in a fertilized grassland in the Loess Plateau. We focused on one important grassland ecosystem function—productivity. Specifically, we studied the following: (1) the relationship between the functional traits and relative biomass of different species under fertilization. (2) The relationship between community-level functional traits and community AGB. (3) The effects of CWM traits on grassland productivity.

## 2. Results

### 2.1. Principal Component Analysis of Functional Traits and Relative Biomass

The relative biomass was used to represent the dominance degree of the six main species. We found that the addition of N and P resulted in various relative biomasses of different species (Appendix A). The effect of N addition was significant for relative biomass of *Lespedeza*
*davurica*, *Artemisia*
*sacrorum*, and *Artemisia scoparia*. P addition significantly affected relative biomass of all species except *A. sacrorum* and *Potentilla*
*tanacetifolia*. There was significant interaction between N and P for relative biomass of *Bothriochloa ischaemum*, *L. davurica*, and *A. sacrorum* (Appendix A).

Different relationships between functional traits and relative biomass were recorded among species following the addition of different levels of N and P (Figure 1). Increased relative biomass of *B. ischaemum* was positively correlated with LN, H_max_, and LN/P under N addition alone and N100 combined with P addition. Increased relative biomass of *L. davurica* was positively correlated with LP under P addition alone. Increased relative biomass of *Stipa bungeana* was positively correlated with LN and LN/P under N addition alone. Increased relative biomass of *A. sacrorum* was positively correlated with H_max_ under N100 combined with P addition. Increased relative biomass of *P. tanacetifolia* was positively correlated with LTD and LDMC under N25 combined with P addition. Increased relative biomass of *A. scoparia* was positively correlated with LN, H_max_, and SLA under N100 combined with P addition (Figure 1; Appendix A).

### 2.2. The Relationship between Community−Weighted Mean Traits and Community Aboveground Biomass

The community AGB was significantly positively correlated with CWM_H_max_ (*r* = 0.80, *p* < 0.001), CWM_SLA (*r* = 0.54, *p* < 0.001), and CWM_LN (*r* = 0.49, *p* < 0.001), while there was a significantly negative correlation with CWM_LDMC (*r* = −0.54, *p* < 0.001) and CWM_LTD (*r* = −0.53, *p* < 0.001; Figure 2). The CWM_H_max_ was significantly positively correlated with CWM_SLA (*r* = 0.64, *p* < 0.01), whereas it was significantly negatively correlated with CWM_LTD (*r* = −0.62, *p* < 0.001) and CWM_LDMC (*r* = −0.60, *p* < 0.001). The CWM_LDMC had a significantly positive correlation with CWM_LTD (*r* = 0.67, *p* < 0.001), while a significantly negative correlation with CWM_SLA (*r* = −0.68, *p* < 0.001). The CWM_SLA had a significantly positive correlation with CWM_LN (*r* = 0.31, *p* < 0.05) and CWM_LP (*r* = 0.32, *p* < 0.01; Figure 2).

### 2.3. Principal Component Analysis of Community−Weighted Mean Traits

Principal component analysis (PCA) compressed the variation of CWM traits into two principal components, representing 73.4% of the total variation (Figure 3A). The first principal component (CWM1) captured 51.1% of the total variation, exhibiting positive correlations with CWM_SLA and CWM_H_max_, but negative correlations with CWM_LTD and CWM_LDMC. The contribution of CWM_SLA and CWM_H_max_ to CWM1 variables was 23.5% and 15.0%, respectively. The contribution of CWM_LTD and CWM_LDMC to CWM1 variables was 21.9% and 17.9%, respectively (Figure 3B).

The second principal component (CWM2), which accounted for 22.3% of the total information, exhibited positive correlations with CWM_LN/P and negative correlations with CWM_LP (Figure 3A). The contribution of CWM_LN/P to CWM2 variables was 37.7%. The contribution of CWM_LP to CWM2 variables was 41.2%. (Figure 3C).

### 2.4. Influence Factors of Grassland Productivity

The structural equation modeling (SEM) explained significant variation in productivity (*p* = 0.996) (Figure 4). The explanatory variables explained 75% of the variation in productivity. Productivity was positively influenced by CWM1 and CWM2 (the standardized path coefficients: *r* = 0.34, *p* < 0.001 and *r* = 0.18, *p* < 0.05, respectively). Additionally, N and P addition increased CWM1 (*r* = 0.45, *p* < 0.001 and *r* = 0.41, *p* < 0.05, respectively), and N addition increased CWM2 (*r* = 0.57, *p* < 0.001), while P addition decreased CWM2 (*r* = −0.65, *p* < 0.001; Figure 4).

## 3. Discussion

Grasslands in the Loess Plateau play a key role in offering ecosystem service and function [29]. Our study highlights the important role of CWM traits for predicting grassland productivity in the Loess Plateau. In this study, we explained the change in grassland productivity from the perspective of plant functional traits, suggesting that species with specific traits such as a high competitive ability or high nutrient use efficiency may be the important drivers in productivity. Studying the relationship between nutrient-induced variations in grassland productivity and plant functional traits was important for grassland restoration and management.

For these dominant species, we evaluated the linkages between plant functional traits and relative biomass of each species under different nutrient levels. Our results showed that different species exhibit different relationships between functional traits and relative biomass under different nutrient addition levels (Figure 1). Different linkages between functional traits and relative biomass may reflect different adaptation strategies to nutrient addition. For *L.davurica*, aboveground biomass increased under P addition alone, and was positively correlated with LP. This may because *L.davurica* is a legume plant with N-fixing function, and P limiting lifted as a result of the P addition. Moreover, a high level N addition may break symbiotic N fixation [32], thus leading to aboveground biomass increasing under only P addition. Accordingly, P fertilization was supposed to be considered when using fertilization to promote grassland restoration in the Loess Plateau.

The mass ratio hypothesis describes the advantage of traits in the community [20]. We found that community AGB was closely associated to CWM traits. This result indicates that the mass ratio hypothesis plays a role in the grassland of the Loess Plateau. The dominant effect of the leaf nutrient-use strategies on productivity may be related to the leaf economics spectrum [33]. Explorative plant species generally have higher SLA and LN and lower LDMC than conservative plant species [34]. In this study, the positive association of CWM_LN, CWM_SLA, and CWM_H_max_, and the negative association of CWM_LTD and CWM_LDMC with community AGB implies acquisitive and conservative strategies across the studied species within communities. Studies have shown that the competition of plants for nutrient and light increased following fertilization [25,35]. The dominant species could maintain high grassland aboveground biomass through the traits related to competition ability. LN is closely correlated with leaf growth and defense strategies [36]. A high SLA tends to lead to high community AGB, and SLA is closely related to the relative growth rate and is a good predictor of plant responses to resource availability [37]. The positive relationship between CWM_H_max_ and community AGB indicates that high community AGB might be associated with tall or fast growing species (Figure 2). Previous studies have also shown that SLA, LDMC, and LN were correlated with aboveground biomass [38]. We also found that community-level traits showed stronger relationships with aboveground biomass than species-level traits. This may because community-level traits could better reflect the community structure [39].

Moreover, we used two PCA axes to obtain important information of community-level traits. CWM1 is positively correlated with CWM_SLA and CWM_H_max_, but negatively correlated with CWM_LTD and CWM_LDMC. CWM2 is positively correlated with CWM_LN/P, but negatively correlated with CWM_LP (Figure 3; Appendix A). In view of this, CWM1 and CWM2 represented the gradients of functional trait composition from species with slow growth and conservative resource use strategies to species with fast growth and acquisitive resource use strategies [39,40,41]. We found that grassland productivity increased with the increase in CWM1 and CWM2 (Figure 4; Appendix A), which indicated that the mass-ratio effect plays a significant role. This is consistent with previous studies [42,43]. Species with acquisitive characteristics would have larger individual aboveground biomass, and thus increased community productivity [21]. For example, high SLA and fast nutrient acquisition would be conducive to fast growth and high productivity [34]. Plant functional traits are considered to be key drivers of ecosystem function [44]. Our results verified this relationship. These findings further prove the close relationship between plant functional traits and productivity, and emphasize the importance of plant functional traits.

## 4. Materials and Methods

### 4.1. Study Area and Experimental Design

The study area was located in Zhifanggou watershed in the Chinese Loess Plateau (36°42′–36°46′ N, 109°13′–109°16′ E). The altitude ranges from 1010 m to 1431 m. The mean annual rainfall and mean temperature in the study location was 528.8 mm and 8.8 °C, respectively. Monthly and daily precipitation and temperature in 2020 are shown in Figure 5. *Bothriochloa ischaemum*, *Lespedeza davurica*, *Stipa bungeana*, *Artemisia sacrorum*, *Potentilla tanacetifolia*, and *Artemisia scoparia* were the dominant species in this grassland.

In August 2017, we conducted the N and P addition experiment in this grassland. We set up twelve main-plots (4 × 4 m) in a randomized block design with three replicated blocks, and each main-plot was divided into four subplots (2 × 2 m). In each block, four N levels (0, 25, 50, and 100 kg N ha^−1^ yr^−1^) were randomly assigned to four main-plots and four levels of P addition rate (0, 20, 40, and 80 kg P_2_O_5_ ha^−1^ yr^−1^) were randomly assigned to four subplots. For more experimental details, see [35]. We used one 1 × 1 m quadrat for the community survey and one 1 × 1 m quadrat for functional trait measurement in each subplot. The soil properties of this grassland under different N and P additions are shown in Table 1.

### 4.2. Community Survey and Trait Measurements

In the summer of 2020, we investigated species abundance, coverage, and maximum plant height (H_max_) of each species in each 1 × 1 m quadrat [45]. Then, the aboveground parts were cut and brought to the laboratory, and oven-dried for 48 h (80 °C) to obtain the AGB of each species. We used the sum of the AGB of all species within this quadrat as a surrogate for productivity in this study.

We measured the plant functional traits of the most dominant species. All functional traits were measured from two to three individuals of each species. Leaf functional traits were measured according to standard methods [46]. SLA (cm^2^ g^−1^) = leaf area/leaf dry mass. Leaf tissue density (LTD; g cm^−3^) = dry mass/leaf volume. LDMC (mg g^−1^) = dry mass/fresh mass. LN (g kg^−1^) was digested with sulfuric acid and determined using a Kjeldahl instrument (Kjektec System 2300 Distilling Unit, Foss, Tecator AB, Hoganas, Sweden), and leaf P content (LP; g kg^−1^) was measured with a molybdenum antimony anti-colorimetric spectrophotometer (UV-2600 spectrophotometer, Shimadzu, Kyoto, Japan). The leaf N/P ratio (LN/P) was calculated as LN divided by LP [46]. CWM traits were calculated as the average of trait values weighted by the AGB of each species within a community [47].

### 4.3. Data Calculation and Analysis

Principal component analysis (PCA) was used to analyze the relationship between the relative biomass and functional traits of the six species under different nutrient levels and to compress the variation in CWM traits into a few principal components using R 4.1.3 (R Development Core Team) using the corrplot package, retaining the first two components (which describe most of the total variance) in the ensuing analysis. The relationship heat map between CWM traits and community AGB was produced in R 4.1.3 (R Development Core Team) using the corrplot package. Structural equation modeling (SEM) was used to explore how CWM traits influence productivity using IBM AMOS version 24.0 (Amos Development Co., Armonk, NY, USA).

## 5. Conclusions

Assessing how plant functional traits influence ecosystem function is important for understanding ecological processes. Our results indicate that dominant species traits could predict ecosystem functioning (productivity) in the Loess Plateau grassland. Thus, the mass ratio hypothesis is proved. Communities dominated by species with fast-growing acquisitive strategies have high productivity in the Loess Plateau grassland. The main novelty of this study is investigating the effect of functional traits after fertilization by adding N and P instead of just N. Overall, our study has strengthened the understanding of mechanisms affecting productivity in the Loess Plateau, and could help predict semiarid grassland responses to future environment change. Future studies need to further clarify the relationship between more species traits and ecosystem function to better understand the effect of fertilization on grassland. 

## Figures and Tables

**Figure 1 plants-11-02045-f001:**
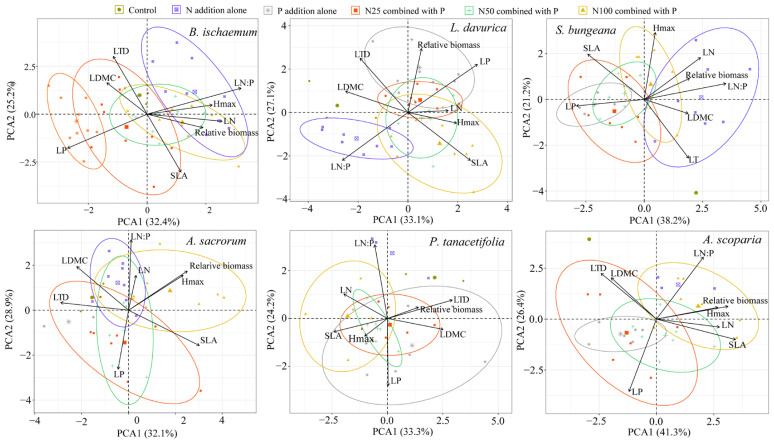
Principal components analysis (PCA) of the relationship between the functional traits and relative biomass of each species under different N and P levels. LN, leaf N content; LP, leaf P content; LN/P, leaf N/P ratio; SLA, specific leaf area; LTD, leaf tissue density; LDMC, leaf dry matter content; H_max_, maximum plant height; RB, relative biomass.

**Figure 2 plants-11-02045-f002:**
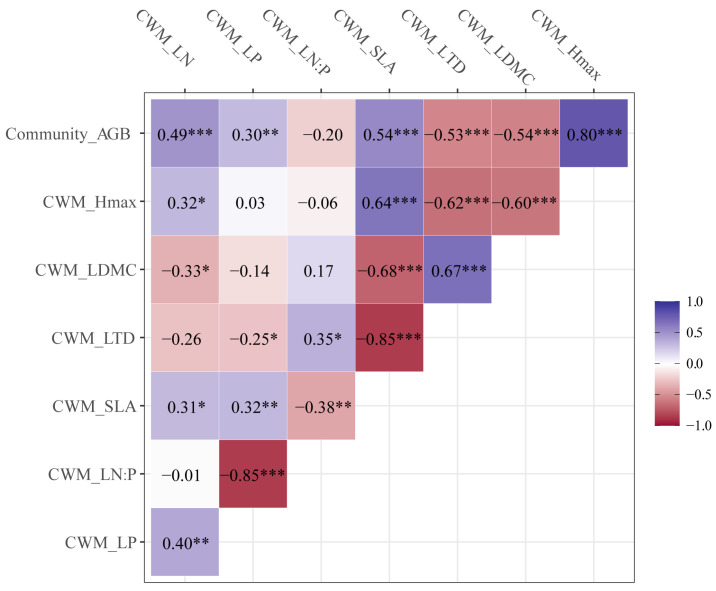
Correlation coefficients of community AGB and community level functional traits. * *p* < 0.05; ** *p* < 0.01; *** *p* < 0.001; LN, leaf N content; LP, leaf P content; LN/P, leaf N/P ratio; SLA, specific leaf area; LTD, leaf tissue density; LDMC, leaf dry matter content; H_max_, maximum plant height.

**Figure 3 plants-11-02045-f003:**
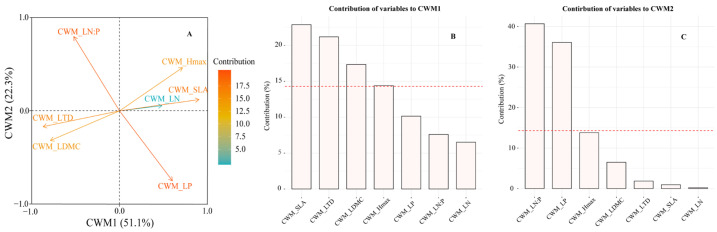
The principal component analysis (PCA) for community-weighted mean (CWM) traits (**A**). The contribution of each community-level trait to CWM1 (**B**) and CWM2 variables (**C**). The red dotted line on bar charts indicates the mean contribution of all seven community-level traits to CWM1 and CWM2 variables. LN, leaf N content; LP, leaf P content; LN/P, leaf N/P ratio; SLA, specific leaf area; LTD, leaf tissue density; LDMC, leaf dry matter content; H_max_, maximum plant height.

**Figure 4 plants-11-02045-f004:**
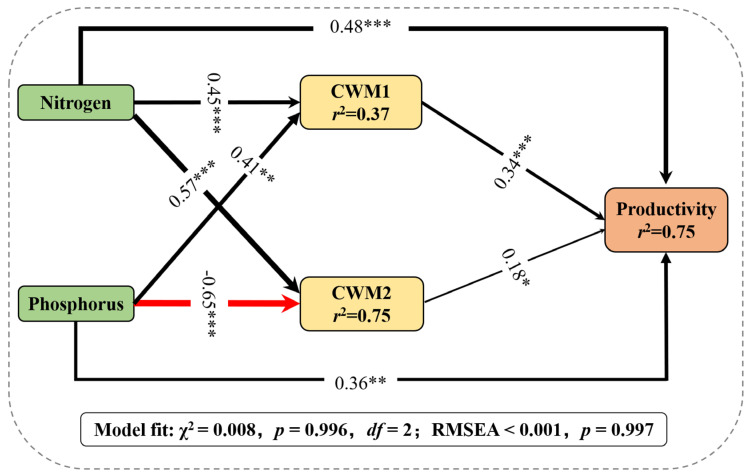
Results of the structural equation model. The arrows represent the hypothesized causal relationships between the variables. The solid lines represent significant relationships (*p* < 0.05). Black arrows represent positive effects, and the red arrows represent negative effects. The values next to the arrows are the standardized path coefficients. The line thickness is proportional to the standardized path coefficient. * *p* < 0.05; ** *p* < 0.01; *** *p* < 0.001. CWM1 and CWM2: the first two PCA axes of CWM traits.

**Figure 5 plants-11-02045-f005:**
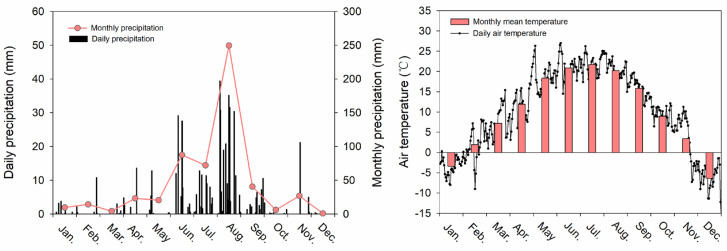
Monthly and daily precipitation and temperature in 2020.

**Table 1 plants-11-02045-t001:** Soil total nitrogen (STN) and soil total phosphorus (STP) of the grassland under different N and P additions (mean ± s.e.; *n* = 3). * *p* < 0.05; ** *p* < 0.01; *** *p* < 0.001.

Soil Properties	Treatments	P0	P20	40	P80
STN (g kg^−1^)	N0	0.54 ± 0.02	0.57 ± 0.02	0.60 ± 0.02	0.53 ± 0.05
	N25	0.65 ± 0.01	0.61 ± 0.01	0.64 ± 0.03	0.53 ± 0.02
	N50	0.70 ± 0.02	0.65 ± 0.02	0.64 ± 0.04	0.64 ± 0.06
	N100	0.74 ± 0.04	0.63 ± 0.03	0.76 ± 0.03	0.67 ± 0.06
STP (g kg^−1^)	N0	0.52 ± 0.01	0.56 ± 0.01	0.60 ± 0.01	0.66 ± 0.02
	N25	0.52 ± 0.01	0.58 ± 0.01	0.61 ± 0.02	0.65 ± 0.01
	N50	0.52 ± 0.02	0.57 ± 0.01	0.57 ± 0.00	0.67 ± 0.02
	N100	0.52 ± 0.01	0.56 ± 0.01	0.58 ± 0.03	0.64 ± 0.02
STN		N ***(0.05)		P *(0.05)	N × P: ns
STP		N: ns		P ***(0.03)	N × P: ns

## Data Availability

All available data can be obtained by contacting the corresponding author.

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
