# Peer review of "Using Trait-Based Methods to Study the Response of Grassland to Fertilization in the Grassland in Semiarid Areas in the Loess Plateau of China"

_plants, 2022, doi:10.3390/plants11152045_

Round 1
Reviewer 1 Report
The manuscript has the following objectives:
- To analyze the relationship between functional traits and relative biomass of different species under fertilization with N and P .
- To investigate the relationship between community level functional traits and community AGB.
- to quantify the effects of CWM traits on grassland productivity.
The manuscript provides the proper context and presents the objectives with adequate precision.
The experimental procedures, design and analysis are adequate to the proposed objectives, as well as the Discussion and Conclusion Sections.
Although they are few, my biggest concerns are:
- Improve the description of the dominance degree of the main species reported in the study. This is a critical point and needs to be clarified in the manuscript. To the reader, there is no clearer information about this.
- The overall quality of figures 1 and 3 is extremely low. A better presentation of these figures is needed.
In the attached file follows the details of my review.

Author Response
Authors’ responses to reviewer 1’ comments on plants-1834213
The manuscript has the following objectives:
- To analyze the relationship between functional traits and relative biomass of different species under fertilization with N and P.
- To investigate the relationship between community level functional traits and community AGB.
- to quantify the effects of CWM traits on grassland productivity.
The manuscript provides the proper context and presents the objectives with adequate precision.
The experimental procedures, design and analysis are adequate to the proposed objectives, as well as the Discussion and Conclusion Sections. Although they are few, my biggest concerns are:
Authors’ response: Many thanks for the constructive comments. The revisions of Reviewer 1 have been highlighted in yellow in the manuscript. Detailed specific explanations were given as following:
Detailed comments:
- Improve the description of the dominance degree of the main species reported in the study. This is a critical point and needs to be clarified in the manuscript. To the reader, there is no clearer information about this.
Authors’ response: Many thanks for the suggestions. The relative biomass was used to represent the dominance degree of the main species reported in the study. We have inserted a table to describe the relative biomass (Table S1). The detailed text about the Table S1 was added in lines 93-99: “The relative biomass was used to represent the dominance degree of the six main species. We found that the addition of N and P resulted in various relative biomass of different species (Table S1). The effect of N addition was significant for relative biomass of Lespedeza davurica, Artemisia sacrorum and Artemisia scoparia. P addition significantly affected relative biomass of all species except A. sacrorum and Potentilla tanacetifolia. There was significant interaction between N and P for relative biomass of B. ischaemum, L. davurica and A. sacrorum (Table S1).”
- The overall quality of figures 1 and 3 is extremely low. A better presentation of these figures is needed.
Authors’ response: Many thanks for the advice. We have redrawn and improved the resolution of the figures 1 and 3.
The details of review in the attached file:
Results:
- What is the dominance level (in terms of occurrence/or proportion of biomass) of the recorded species? To clarify. Bothriochloa ischaemum, Lespedeza davurica, Stipa bungeana, Artemisia sacrorum, Potentilla tanacetifolia and Artemisia scoparia.
Authors’ response: Many thanks for the suggestion. The dominance level of the recorded species has described in Table S1. And we added full name of species Latin name “Bothriochloa ischaemum, Lespedeza davurica, Stipa bungeana, Artemisia sacrorum, Potentilla tanacetifolia and Artemisia scoparia” when they first appeared (lines 92-105).
- Replace ". " with ";".
Authors’ response: Revised accordingly (Lines 142-143).
- I think it's better to invert the colors: Red: negative correlation. Blue: positive correlation.
Authors’ response: Many thanks for the suggestions. We have inverted the colors of Figure 2, and used “Red” to represent negative correlation and “Blue” to represent positive correlation.
- Replace ". " with ";".
Authors’ response: Revised accordingly (Line 166).
- Figure 3: What does the red dotted line on bar charts mean? To specify.
Authors’ response: Many thanks for the advice. The red dotted line on bar charts indicates the mean contribution of all seven community-level traits to CWM1 and CWM2 variables. We added the statement “The red dotted line on bar charts indicates the mean contribution of all seven community-level traits to CWM1 and CWM2 variables” in the Figure 3 legend in lines 182-183.
- Figure 3: It is difficult to read and understand the figure. Resolution and formatting need improve
Authors’ response: Many thanks for the advice. We have improved resolution and formatting of Figure 3 (300 dpi) and rewritten the legend of Figure 3 in lines 181-184: The principal component analysis (PCA) for community-weighted mean (CWM) traits (A). The contribution of each community-level trait to CWM1 (B) and CWM2 variables (C). The red dotted line on bar charts indicates the mean contribution of all seven community-level traits to CWM1 and CWM2 variables. LN, leaf N content; LP, leaf P content; LN:P, leaf N:P ratio; SLA, specific leaf area; LTD, leaf tissue density; LDMC, leaf dry matter content; Hmax, maximum plant height.
We also added description of Figure 3 (B) and (C) in the results to better understand it: “The contribution of CWM_SLA and CWM_Hmax to CWM1 variables was 23.5% and 15.0%, respectively. The contribution of CWM_LTD and CWM_LDMC to CWM1 variables was21.9% and 17.9%, respectively (Fig. 3B)” in lines 172-174 and “The contribution of CWM_LN:P to CWM2 variables was 37.7%. The contribution of CWM_LP to CWM2 variables was 41.2%. (Fig. 3C)” in lines 177-178.
- Replace ". " with ";".
Authors’ response: Revised accordingly (Line 184).
Discussion
- "Aboveground" to "aboveground".
Authors’ response: Revised accordingly (Line 215).
Conclusions
- Throughout the manuscript, the characteristics that quantify the dominance of the main species are not described.
Authors’ response: Many thanks for the suggestion. The characteristics that quantify the dominance of the main species in terms of relative biomass are described in Table S1.

Reviewer 2 Report
The results of the manuscript titled "Use trait-based method to study the response of grassland to fertilization in the grassland in semiarid area in the Loess Plateau of China" written by Yuting Yang et al. showed novelty about the grassland ecosystem productivity under environmental change. It is now very important to carry out such experiments. I highlight that the experiment was conducted in field conditions. Nevertheless, I have some well-meaning comments about the manuscript that could increase its quality and develop some ideas:
1. All used material, machines and devices must be specified (production name, company name, city, country of origin).
2. All used methods must be specified and supported by citations.
3. In the case of a field experiment, the results may be significantly affected by the interaction of the weather conditions and soil properties. In Material and Methods chapter, I kindly ask the authors for a detailed course of weather (temperature and precipitation) of individual months of the experimental years. In the manuscript, this can be supplemented by graphs (course of weather) or tables (soil properties) to help understand some of the differences in results achieved. Such information is very important in plant nutrition experiments.
4. I recommend placing the Materials and Methods chapter before the Results chapter, as is standard practice.
5. In the results, I recommend increasing the resolution of the images for better readability.
Author Response
Authors’ responses to reviewers 2’ comments on plants-1834213
The results of the manuscript titled "Use trait-based method to study the response of grassland to fertilization in the grassland in semiarid area in the Loess Plateau of China" written by Yuting Yang et al. showed novelty about the grassland ecosystem productivity under environmental change. It is now very important to carry out such experiments. I highlight that the experiment was conducted in field conditions. Nevertheless, I have some well-meaning comments about the manuscript that could increase its quality and develop some ideas:
Authors’ response: Many thanks for the constructive comments. The revisions of Reviewer 2 have been highlighted grey in the manuscript. Detailed specific explanations were given as following:
- All used material, machines and devices must be specified (production name, company name, city, country of origin).
Authors’ response: Many thanks for the suggestion. We have added production name, company name, city, country of origin of all used material, machines and devices: “Kjeldahl instrument (Kjektec System 2300 Distilling Unit, Foss, Tecator AB, Hoganas, Sweden)” in lines 294-295, “molybdenum antimony anti-colorimetric spectrophotometer (UV-2600 spectrophotometer, Shimadzu Japan)” in line 296 and “IBM AMOS version 24.0 (Amos Development Co., Armonk, USA)” in lines 308-309.
- All used methods must be specified and supported by citations.
Authors’ response: Many thanks for the suggestion. We added reference No. 46 to support “we investigated species abundance, coverage and maximum plant height (Hmax) of each species in each 1 × 1 m quadrat” in line 285 and reference No. 47 to support “LN (g kg-1) was digested with sulfuric acid and determined using a Kjeldahl instrument (Kjektec System 2300 Distilling Unit, Foss, Tecator AB, Hoganas, Sweden), and leaf P content (LP; g kg-1) was measured with a molybdenum antimony anti-colorimetric spectrophotometer (UV-2600 spectrophotometer, Shimadzu Japan). Leaf N:P ratio (LN:P) was calculated as LN divided by LP” in lines 293-297. We also carefully checked and ensured all used methods were specified and supported by citations.
The added reference was:
[46] Jing, Z.B., Cheng, J.M., Su, J.S., Bai, Y., Jin, J.W., 2014. Changes in plant community composition and soil properties under 3-decade grazing exclusion in semiarid grassland. Ecol. Eng. 64, 171–178.
[47] Pérez-Harguindeguy, N.; Díaz, S.; Garnier, E.; Lavorel, S.; Poorter, H.; Jaureguiberry, P.; Bret-Harte, M.S.; Cornwell, W.K.; Craine, J.M.; Gurvich, D.E.; Urcelay, C.; Veneklaas, E.J.; Reich, P.B.; Poorter, L.; Wright, I.J.; Ray, P.; Enrico, L.; Pausas, J.G.; de Vos, A.C.; Buchmann, N.; Funes, G.; Quetier, F.; Hodgson, J.G.; Thompson, K.; Morgan, H.D.; ter Steege, H.; van der Heijden, M.G.A.; Sack, L.; Blonder, B.; Poschlod, P.; Vaieretti, M.V.; Conti, G.; Staver, A.C.; Aquino, S.; Cornelissen, J.H.C. New handbook for standardised measurement of plant functional traits worldwide. Aust. J Bot. 2013, 61, 167–234.
- In the case of a field experiment, the results may be significantly affected by the interaction of the weather conditions and soil properties. In Material and Methods chapter, I kindly ask the authors for a detailed course of weather (temperature and precipitation) of individual months of the experimental years. In the manuscript, this can be supplemented by graphs (course of weather) or tables (soil properties) to help understand some of the differences in results achieved. Such information is very important in plant nutrition experiments.
Authors’ response: Many thanks for the suggestion. We have used figure 5 to show a detailed course of monthly and daily precipitation and temperature in 2020, and table 1 to show soil total nitrogen (STN) and soil total phosphorus (STP) of the grassland under different N and P additions.
- I recommend placing the Materials and Methods chapter before the Results chapter, as is standard practice.
Authors’ response: Many thanks for the advice. We used the Plants Microsoft Word template file, which placed the Materials and Methods chapter after the Results chapter.
- In the results, I recommend increasing the resolution of the images for better readability.
Authors’ response: Many thanks for the advice. Sorry for the lower resolution of the images. We have improved the resolution of all the images in the manuscript.

Reviewer 3 Report
Manuscript is dealing with relevant parameters that are used for describing of ecosystem mainly in community ecology. Relation with N and P with aboveground plant parts are described in the literature enough, also used parameters are known more years in ecology studies. However, focusing the one kind of ecosystem to these parameters and elements can be novelty of this research. Connection of used parameters (grassland productivity, adding nutrients in the form of nitrogen and phosphorous, focusing to aboveground biomass, etc.) with PCA analyses is good way to recognize maybe the most relevant tested parameters that can be helpful to understand (and later maybe improving) the processes in the ecosystem. Here mainly for grassland in semiarid area.
I have following recommendations for increasing of manuscript quality:
- I highly recommend to insert table with data that was used for statistical evaluation and are from articles No. 31, 32, 33. Without these data is very hardly to see importance of authors research and novelty in these statistics findings.
- Graphs in this form are not readable - I have big problem to read each text on Fig. 1 and 3.
- Not logical numbering of citations - after 32 is continuing with citation No. 36. Please, renumbered it.
- maybe it is not important "doubled" information from the Fig. 2 in the text - authors should be focused in the most relevant dependences founded in their result, not mentioned everything what is seeing on Fig. 2 (subchapter 2.2)
Author Response
Authors’ responses to reviewer 3’ comments on plants-1834213
Manuscript is dealing with relevant parameters that are used for describing of ecosystem mainly in community ecology. Relation with N and P with aboveground plant parts are described in the literature enough, also used parameters are known more years in ecology studies. However, focusing the one kind of ecosystem to these parameters and elements can be novelty of this research. Connection of used parameters (grassland productivity, adding nutrients in the form of nitrogen and phosphorous, focusing to aboveground biomass, etc.) with PCA analyses is good way to recognize maybe the most relevant tested parameters that can be helpful to understand (and later maybe improving) the processes in the ecosystem. Here mainly for grassland in semiarid area. I have following recommendations for increasing of manuscript quality:
Authors’ response: Many thanks for the constructive comments. The revisions of Reviewer 3 have been highlighted in green in the manuscript. Detailed specific explanations were given as following:
Detailed comments:
- I highly recommend to insert table with data that was used for statistical evaluation and are from articles No. 31, 32, 33. Without these data is very hardly to see importance of authors research and novelty in these statistics findings.
Authors’ response: Many thanks for the suggestion. We have insert tables that was used for statistical evaluation, including relative biomass of the six species (Table S1), soil total nitrogen (STN) and soil total phosphorus (STP) of the grassland (Table 1) and community weighted leaf traits, maximum plant height (Hmax) and community aboveground biomass (Table S2) under different N and P addition additions. We also added table: the analysis of variance results (F values) for the effects of N addition (N), P addition (P), species and their interactions on leaf trait and maximum plant height (Hmax) (Table S3).
We have reordered and renumbered the logical numbering of citations in the text, and the reference No. 33 was revised as No. 36. We carefully read articles No. 31, 32, 36 and provided relevant data as following. The references No. 31 and 32 were used to supported: “However, the grassland productivity here are usually limited by both N and P in the Loess Plateau” in line 76. The reference No. 31 stated that: Mean soil total nitrogen (STN) and soil total phosphorus (STP) of grassland on the Loess Plateau were 0.34 kg m−2 and 0.31 kg m−2, respectively, which were lower than the mean STN and STP in the whole of China. Thus, the Loess Plateau region had a relatively low level of STN and STP in China. The reference No. 32 stated that: The soil of farming-withdrawn grassland in this study has low available N (50 mg kg−1), low available P (1.7 mg kg−1), which were obviously lower than that of native vegetation (available N 109.5 mg kg–1, available P: 3.5 mg kg–1).
The reference No. 36 were used to supported: “Researches showed that the competition of plants for nutrient and light increased following fertilization” in line 232. The reference No. 36 stated that: The soil surface light availability in this grassland community reduced to 15%~40% following fertilization. Thus it increased light competition.
- Graphs in this form are not readable - I have big problem to read each text on Fig. 1 and 3.
Authors’ response: Sorry for the unclear statement. We have improved resolution of Fig. 1 and 3 (300 dpi) and rewritten the text on Fig. 1 and 3 in the results.
Fig.1 now read as “Different relationships between functional traits and relative biomass were recorded among species following addition of different levels of N and P (Fig. 1). Increased relative biomass of B. ischaemum was positively correlated with LN, Hmax and LN:P under N addition alone and N100 combined with P addition. Increased relative biomass of L. davurica was positively correlated with LP under P addition alone. Increased relative biomass of S. bungeana was positively correlated with LN and LN:P under N addition alone. Increased relative biomass of A. sacrorum was positively correlated with Hmax under N100 combined with P addition. Increased relative biomass of P. tanacetifolia was positively correlated with LTD and LDMC under N25 combined with P addition. Increased relative biomass of A. scoparia was positively correlated with LN, Hmax and SLA under N100 combined with P addition (Fig. 1; Table S1 and Table S2).” in lines 100-110.
We added “The contribution of CWM_SLA and CWM_Hmax to CWM1 variables was 23.5% and 15.0%, respectively. The contribution of CWM_LTD and CWM_LDMC to CWM1 variables was 21.9% and 17.9%, respectively.” in lines 172-174 to describe Fig.3B and “The contribution of CWM_LN:P to CWM2 variables was 37.7%. The contribution of CWM_LP to CWM2 variables was 41.2%” in lines 177-178 to describe Fig.3C.
We rewritten the legend of Figure 3 in line 181-184: The principal component analysis (PCA) for community-weighted mean (CWM) traits (A). The contribution of each community-level trait to CWM1 (B) and CWM2 variables (C). The red dotted line on bar charts indicates the mean contribution of all seven community-level traits to CWM1 and CWM2 variables. LN, leaf N content; LP, leaf P content; LN:P, leaf N:P ratio; SLA, specific leaf area; LTD, leaf tissue density; LDMC, leaf dry matter content; Hmax, maximum plant height.
- Not logical numbering of citations - after 32 is continuing with citation No. 36. Please, renumbered it.
Authors’ response: Many thanks for the advice. We have reordered and renumbered the logical numbering of citations in the text. The order of citations and references in the text has been modified correspondingly.
- maybe it is not important "doubled" information from the Fig. 2 in the text - authors should be focused in the most relevant dependences founded in their result, not mentioned everything what is seeing on Fig. 2 (subchapter 2.2)
Authors’ response: Many thanks for the suggestion. We have simplified the text on Fig. 2 and focused in the most relevant dependences founded in our result. The text has been revised as “The community AGB was significantly positively correlated with CWM_Hmax (r=0.80, p < 0.001), CWM_SLA (r=0.54, p < 0.001) and CWM_LN (r=0.49, p < 0.001), while there had a significantly negative correlation with CWM_LDMC (r=-0.54, p < 0.001) and CWM_LTD (r=-0.53, p < 0.001; Fig. 2). The CWM_Hmax was significantly positively correlated with CWM_SLA (r=0.64, p < 0.01), whereas significantly negatively correlated with CWM_LTD (r=-0.62, p < 0.001) and CWM_LDMC (r=-0.60, p < 0.001). The CWM_LDMC had a significantly positive correlation with CWM_LTD (r=0.67, p < 0.001), while a significantly negative correlation with CWM_SLA (r=-0.68, p < 0.001). The CWM_SLA had a significantly positive correlation with CWM_LN (r=0.31, p < 0.05) and CWM_LP (r=0.32, p < 0.01; Fig. 2)” in lines 146-161.

Round 2
Reviewer 3 Report
Improved changes were included properly and as was recommended. Figures are readable, now, and necessary tables with data are inserted. I can recommend this article for publishing in Plants journal.